# Drug-Resistant *Helicobacter pylori*: Diagnosis and Evidence-Based Approach

**DOI:** 10.3390/diagnostics13182944

**Published:** 2023-09-14

**Authors:** Vaneet Jearth, Mitali Madhumita Rath, Abhirup Chatterjee, Aditya Kale, Manas Kumar Panigrahi

**Affiliations:** 1Post Graduate Institute Medical Education and Research, Chandigarh 160012, India; vaneet.jearth@gmail.com (V.J.); drabhirup@gmail.com (A.C.); 2Hi-Tech Medical College and Hospital, Bhubaneswar 751010, India; mitali.m.rath@gmail.com; 3Tata Memorial Hospital, Mumbai 400012, India; adityapkale@yahoo.com; 4All India Institute of Medical Sciences, Bhubaneswar 751019, India

**Keywords:** antibiotic resistance, single-drug resistance, multidrug resistance, heteroresistance, antibiotic susceptibility testing, treatment failure, gastric cancer, eradication failure, diagnosis

## Abstract

*Helicobacter pylori* (*H. pylori*) is the most common chronic bacterial infection, affecting approximately half of the world’s population. *H. pylori* is a Class I carcinogen according to the World Health Organization, and the International Agency for Research on Cancer (IARC) has linked it to 90% of stomach cancer cases worldwide. The overall pattern points to a yearly reduction in eradication rates of *H. pylori* with the likelihood of success further decreasing after each unsuccessful therapeutic effort. Antimicrobial resistance in *Helicobacter pylori* is a major public health concern and is a predominant cause attributed to eradication failure. As a result, determining *H. pylori*’s antibiotic susceptibility prior to the administration of eradication regimens becomes increasingly critical. Detecting *H. pylori* and its antimicrobial resistance has traditionally been accomplished by time-consuming culture and phenotypic drug susceptibility testing. The resistance of *H. pylori* to different antibiotics is caused by various molecular mechanisms, and advances in sequencing technology have greatly facilitated the testing of antibiotic susceptibility to *H. pylori*. This review will summarize *H. pylori* antibiotic resistance patterns, mechanisms, and clinical implications. We will also review the pros and cons of current antibiotic susceptibility testing methods. Along with a comparison of tailored susceptibility-guided regimens and empirical therapy based on the latest evidence, an evidence-based approach to such situations will be explored.

## 1. Introduction

*Helicobacter pylori* (*H. pylori*) is the most prevalent chronic bacterial infection, affecting an estimated 50 percent of the world’s population. Despite being asymptomatic in 80–90% of infected people, chronic infection can result in gastritis and peptic ulcer disease and may potentially advance to gastric adenocarcinoma and mucosa-associated lymphoid tissue (MALT) lymphoma [1,2,3]. The World Health Organization has classified *H. pylori* as a Class I carcinogen, and the International Agency for Research on Cancer (IARC) has implicated it in 90% of gastric cancers globally [1,4]. Gastric cancer is a leading cause of morbidity and mortality worldwide, ranking fifth among the most prevalent cancers and third in terms of mortality rate [5,6]. Several studies, including a recent updated meta-analysis, indicate that *H. pylori* eradication can reduce the incidence of gastric cancer in infected individuals and patients with gastric neoplasia [7,8,9]. Elimination of *H. pylori* at a young age or in patients with early-stage infection is the most cost-effective method for preventing gastric cancer [1,7,9]. Additionally, eradication reduces the recurrence of gastric and duodenal ulcers and resolves two-thirds of MALT lymphoma [10]. Therefore, all major society guidelines recommend *H. pylori* eradication in infected individuals [1].

At present, the primary obstacle to the eradication of *H. pylori* infection is the emergence of antibiotic resistance, which significantly compromises the effectiveness of eradication regimens [5]. The global trend indicates a gradual decline in eradication rates every year, with the probability of success diminishing even further after each unsuccessful therapeutic endeavor [11]. The ramifications of such failures include repeated exposure to high-dose acid suppression and various antibiotics, as well as the evolution of antibiotic resistance. There are also healthcare costs connected with such failures due to accompanying gastroduodenal disorders. However, it must be emphasized that, apart from the predominant reason of increasing antibiotic resistance, reasons for treatment failure may be complex, multidimensional, and associated with numerous other factors (Figure 1) [7,11,12]. Knowledge of these factors also aids in optimizing eradication rates, which have dropped to between 50 and 75 percent in some nations [12]. Currently, an acceptable *H. pylori* treatment regimen is one that yields at least a 90% cure rate, while it has been recommended that an optimized regimen should attain 95% cure rates on a consistent basis [13]. The current treatment regimens recommended by international guidelines are mostly empirical in nature. As a result of diminishing eradication rates, bismuth-based quadruple therapy is being increasingly advocated as a first-line treatment option, replacing clarithromycin-based triple treatments, and has been found to achieve an over 90% success rate [14].

This review will provide a summary of current *H. pylori* antibiotic resistance patterns, resistance mechanisms, and clinical implications. In addition, we will discuss the utilization, advantages, and limitations of different antibiotic susceptibility testing methods that are now in use. In addition, an evidence-based approach to the management of such cases will be investigated, along with a comparison of tailored susceptibility-guided regimens and empirical therapy in accordance with the most recent evidence.

## 2. Global Prevalence of Antibiotic Resistance among *H. pylori*

Several recent systematic reviews and meta-analyses have been conducted to estimate the frequency of antibiotic resistance among *H. pylori* strains using regional and worldwide data [15,16,17,18]. Despite significant regional heterogeneity, an increase in *H. pylori*’s resistance to antibiotics has been reported worldwide, accompanied by a decline in eradication rates. Regional heterogeneity can be explained by varying patterns of antibiotic consumption and disease burdens for which antibiotics are used in respective countries. The therapeutic arsenal for eradicating *H. pylori* consists of a limited number of antibiotics, and the widespread use of antibiotics at the population level to treat other common diseases is a significant contributor to the emergence of increased resistance to various antibiotics. Antibiotic resistance rates for the antibiotics used to treat *H. pylori* infection vary globally, ranging from 15% to 50% [16,19]. Figure 2 summarizes the global prevalence of antibiotic-specific resistance rates by region.

In the most recent systematic review and meta-analysis, the antibiotic resistance pattern of *H. pylori* in Southeast Asian nations was examined by analyzing 23 studies. The authors reported antibiotic resistance prevalence rates of 23%, 27%, 69%, 16%, 34%, and 14% for amoxicillin, clarithromycin, metronidazole, tetracycline, levofloxacin, and furazolidone, respectively. According to subgroup analysis, prevalence was high in Pakistan, India, and Bangladesh. In addition, a ten-year trend analysis revealed an increase in the prevalence of clarithromycin (from 21% to 30%), ciprofloxacin (from 3% to 16%), and tetracycline (from 5% to 20%) resistance from 2003 to 2022 (Figure 3) [15]. Recent large-scale studies from the United States, China, and Europe all agreed on the worrying rise in antibiotic resistance [20,21,22]. Rifabutin resistance appears to be rare, based on limited evidence [19].

## 3. Mechanisms of Drug Resistance and Clinical Implications

These are broadly classified into three types of drug resistance pattern: single-drug resistance (SDR), multidrug resistance (MDR), and heteroresistance (HR).

**SDR:** When resistance develops against a single class of antibiotics, the mechanism is usually specific. This is known as SDR. Indeed, it is class-specific. SDR is typically defined through three mechanisms: (i) drug target-mediated resistance, (ii) drug detoxication, and (iii) drug uptake limitation [23,24]. Figure 4 demonstrates the predominant molecular mechanisms of single-drug resistance in Helicobacter pylori.

Clinical relevance of SDR in terms of efficacy of eradication regimens involving these medications has primarily been examined for clarithromycin, levofloxacin, and metronidazole. As a secondary outcome, Savoldi et al. analyzed data from 45 studies and reported a 7-fold (OR, 6.97; 95% CI, 5.23–9.01; *p =* 0.001) increase in the likelihood of eradication failure in patients with clarithromycin-resistant *H. pylori* strains treated with clarithromycin-containing regimens. A similar strong association was reported for levofloxacin (OR, 8.18; 95% CI, 3.81–17.56; *p =* 0.001) and combined clarithromycin and metronidazole resistance (OR, 9.40; 95% CI, 5.48–16.12; *p =* 0.001), though the clinical impact of metronidazole-only resistance was comparatively less significant (OR, 2.52; 95% CI, 1.82–3.48; *p* = 0.004) [16].

**MDR:** Multidrug resistance is defined as the simultaneous occurrence of resistance to three or more classes of antibiotics. Recent reports have indicated the alarming prevalence of MDR strains with triple resistance to clarithromycin, metronidazole, and quinolones as the most frequently reported MDR pattern. Primary MDR rates ranged from less than 10% in most parts of Europe to more than 20% in India and more than 40% in Peru [25,26]. Several mechanisms for MDR in *H. pylori* have been identified, but their clinical significance is unclear. These mechanisms include (i) increased drug efflux due to the activation of channels and efflux pumps; (ii) decreased drug uptake; (iii) biofilm formation; and (iv) transformation into an antibiotic-resistant coccoid form. Efflux pumps are upregulated or overexpressed, resulting in greater drug efflux from the cell. Biofilms are an extracellular matrix created by organisms as a long-term survival strategy that works as an effective, non-specific antibiotic barrier. They have emerged as an important MDR mechanism. These strains are also thought to have different modes of transmission. *H. pylori* can exist in three states: live bacillary, non-viable degenerative, and quiescent coccoid. To kill these coccoid forms, antibiotics must have a substantially higher minimum inhibitory concentration (MIC). Figure 5 depicts the predominant mechanisms attributable to *H. pylori* multidrug resistance [24].

**HR:** It is the coexistence of susceptible and resistant strains of an organism against a particular antimicrobial agent [27]. Recently, HR has been defined as the presence of a resistant strain in a subpopulation with a MIC that is at least eight times higher than the concentration required to inhibit the growth of the primary population [28]. HR in *H. pylori* can be intra-niche (when susceptible and resistant strains are isolated from the same gastric mucosal location) or inter-niche (when biopsies collected from separate regions of the gastric mucosa reveal *H. pylori* strains with different susceptibility patterns). HR can be monoclonal or polyclonal, and it can exist in both eradication-naive and previously treated patients. Polyclonal HR is characterized by the presence of genetically distinct clones exhibiting a stable antibiotic sensitivity phenotype. Monoclonal HR is characterized by the presence of genetically identical strains that can give rise to new strains with a variable pattern of susceptibility or resistance. Recent research suggests that HR in *H. pylori* is largely caused by co-infection with many strains in developing countries, whereas it is caused by the microevolution of a single *H. pylori* strain in response to antibiotic pressure in industrialized countries. HR is a serious clinical concern that is rarely addressed in major treatment guidelines and may be a precursor to future SDR or MDR [24,29]. According to a recent study by Kocsmár et al., one-fifth of clarithromycin-resistant patients can be missed with a biopsy from only the antrum, and inter-niche-only techniques can lower HR by more than 50%. To counteract this phenomenon, evidence suggests taking multiple biopsies from different gastric regions or evaluating multiple bacterial colonies from the same sample for drug susceptibility testing [30].

## 4. Detection of Antibiotic Resistance

Antimicrobial resistance is a ubiquitous problem requiring various solutions. The increasing prevalence of antibiotic resistance among *H. pylori* isolates worldwide has severely impacted the efficacy of eradication regimens, which could have significant health and economic consequences [31]. In order to personalize *H. pylori* therapy and formulate treatment guidelines for specific populations based on primary resistance patterns, antibiotic susceptibility testing (AST) plays a crucial role. The detection of antibiotic resistance can be assessed either by phenotypic methods employing the growth of *H. pylori* in culture from endoscopically retrieved gastric tissue or by genotypic or molecular methods employing pure culture isolates or biological specimens such as biopsy, stool, or saliva [23].

### 4.1. Phenotypic Methods

These methods involve the cultivation of *H. pylori* through bacterial culture. A bacterial isolate acquired by culturing is used to determine antimicrobial sensitivity. This can be accomplished using either the dilution method (agar or broth dilution method) or the diffusion method (disc or E-test). Methods such as agar dilution, broth dilution, and the E-test are utilized to determine the minimum inhibitory concentration (MIC) of antibiotics. Quantitative approaches include agar dilution and broth dilution [23]. 

The agar dilution method is the gold standard and is recommended by the Clinical and Laboratory Standard Institute (CLSI) [32]. First, 1 to 3 mL of a 2.0 McFarland-adjusted bacterial suspension (containing 1 × 10^7^ to 1 × 10^8^ CFU/mL) is spot inoculated on Mueller–Hinton agar supplemented with 5 to 10% sheep or horse blood and 2-fold dilutions of the antibiotics. Plates are read for bacterial growth after 72 h of incubation at 35 °C in microaerophilic conditions. The maximum dilution of antibiotics that results in no growth is referred to as the MIC. *H. pylori* strains are classified as sensitive, moderately resistant, or resistant based on the concentrations at which they are sensitive. According to the European Committee on Antimicrobial Susceptibility Testing (EUCAST), the MIC breakpoint for amoxicillin is >0.125 mg/L, levofloxacin’s is >1 mg/L, clarithromycin’s is >0.5 mg/L, tetracycline’s is >1 mg/L, metronidazole’s is >0.8 mg/L, and rifampicin’s is >1 mg/L [33].

The disc diffusion method offers a cut-off dependent on the diameter of the zone of inhibition. This method is widely used in epidemiological investigations, but its application in ordinary day-to-day practice has been difficult and time consuming due to tedious preparations, the length of time required, and the fact that it is not cost-effective [23,34,35]. Due to the difficulty of culturing *H. pylori* in broth without defibrinated blood, the broth dilution method is rarely employed [36].

The E-test employs a bacterial solution calibrated to the McFarland standard of 3.0 on Muller–Hinton agar supplemented with 5–10% sheep blood [34]. Under microaerophilic circumstances, an E-test strip impregnated with escalating quantities of antibiotics is placed on inoculation plates and incubated for 72 hours at a temperature of 37 °C. The MIC is provided by an elliptical zone of inhibition. When compared to agar dilution or broth dilution procedures, the E-test is straightforward and easy to execute, and it has been found to correlate closely with gold standard (agar dilution) results, except for metronidazole, where recent investigations have revealed some inconsistencies [37,38,39]. The disc diffusion method is a semiquantitative approach to assessing antibiotic resistance. It measures antimicrobial resistance using cut-off diameters of the zone of inhibition [23].

### 4.2. Molecular Methods

Although *H. pylori* culture followed by MIC-based AST is regarded as the gold standard, the widespread use of culture-based techniques is limited for a number of reasons. *H. pylori*’s finicky nature necessitates well-equipped laboratories with trained personnel to provide supplemental growth conditions, a microaerophilic atmosphere, and up to 14 days of incubation. Culture-based methodologies are also affected by sample processing delays and recent exposure to PPIs. This corresponds to a success rate of between 55 and 73% in routine clinical practice. In addition, the requirement of endoscopy for tissue acquisition restricts their use to endoscopy cohorts. Molecular-based methods have emerged as a more rapid (up to <4 h), cost-effective, and highly reproducible alternative to culture-based methods for detecting *H. pylori* and determining its resistance, given the challenges associated with traditional culture methods. Molecular methods can detect heteroresistant infections and are less affected by the modulation of bacterial burden caused by the recent use of PPIs [23,24,40,41]. Recent large RCTs have demonstrated that molecular testing-guided therapy is comparable to culture-based susceptibility testing-guided therapy in first-line therapy and non-inferior to the latter in third-line treatment of *H. pylori* infection, thereby supporting the use of molecular testing-guided therapy for *H. pylori* eradication [42].

Another benefit of molecular approaches is that gastric samples collected through rapid urease tests (RUTs) can be utilized for PCR-based molecular testing. The observed correlation between the utilization of RUT gastric biopsies for molecular testing after 30 days at room temperature was 93% [43]. In addition, these methods allow for the non-invasive detection of antibiotic resistance [40]. Gastric contents (i.e., gastric fluid, mucus, or mucosal biopsies), formalin-fixed, paraffin-embedded (FFPE) tissue, stools, or repurposed tissue from histology or rapid urease tests can be used for testing. A recent meta-analysis found that genotypic testing of clarithromycin resistance from stool specimens is an accurate, convenient, and rapid detection method with pooled sensitivity and specificity values of 0.93 (95% confidence interval (CI): 0.90 to 0.96) and 0.98 (95% CI: 0.93 to 1.00), respectively [44].

The alterations in genes that cause antibiotic resistance in *H. pylori* have been extensively investigated, particularly for the most routinely used antibiotics in eradication regimens, such as clarithromycin and levofloxacin. These are predominantly attributable to recognized localized mutations such as point mutations in 23S rRNA for clarithromycin resistance and in *gyrA* for levofloxacin resistance (Figure 6). Without conducting an antibiogram, molecular-based methods rely primarily on the detection of these specific *H. pylori* mutations encoding resistance in order to determine genotypic susceptibility to commonly employed antibiotics. The majority of assays are polymerase chain reaction (PCR)-based, and various techniques, including conventional PCR followed by Sanger sequencing, real-time (RT) PCR, droplet digital PCR, DNA microarray technology, multiplex quantitative PCR, and next-generation sequencing (NGS), are employed for this purpose. Moreover, these methodologies can be culture-based or culture-free when applied directly to biological samples [23,24,41].

Using specific primers, conventional PCR amplifies specific regions in target genes known to contain major mutations, as shown in Figure 4 for a particular antibiotic. The amplified primers are then subjected to DNA sequencing using the Sanger method in order to detect mutations. Although this method is regarded as the gold standard for identifying mutations and demonstrates a strong correlation with phenotypic antibiotic resistance [36], it is not cost-effective in routine clinical contexts [45].

The use of RT-PCR assays, which provide results for simultaneous detection of *H. pylori* and antimicrobial resistance in real time within hours and can be performed in any microbiological laboratory [40,41], is a more practical, rapid, and cost-effective alternative. Several commercially available kits utilize this principal for detection of clarithromycin resistance including *H. pylori* ClariRes (Ingenetix, Vienna, Austria) [46], Allplex *H. pylori* and ClariR (Seegene, Republic of Korea) [47], Lightmix *H. pylori* (TIBMolbiol, Germany) [48], *H. pylori* TaqMan real-time PCR assay (Meridian Bioscience, United States) [49], Amplidiag *H. pylori* + ClariR (Mobidiag, Espoo, Finland) [50], RIDA GENE *H. pylori* (r-Biopharm, Darmstadt, Germany) [51], Seeplex ClaR-*H. pylori* ACE detection system (See-gene, Eschborn, Germany), and the MutaREAL *Helicobacter pylori* kit (Immunodiagnostik, Benshiem, Germany) [41,52]. The GenoType HelicoDR assay (Hain Lifescience, Nehren, Germany) uses a combination PCR and DNA hybridization procedure to detect combined clarithromycin and levofloxacin resistance-causing mutations [53]. Despite the fact that these methods have demonstrated excellent performance using a biopsy or stool sample to simultaneously detect infection and antimicrobial resistance, especially for clarithromycin, with a sensitivity and specificity of up to 94% and 100%, respectively, local validation of kits in reference to culture-based AST is recommended [23,41]. The sensitivity of these approaches varies depending on the DNA extraction process and molecular assay used. False negative results can be achieved if paraffin-embedded gastric biopsy samples are used due to DNA fragmentation caused by fixatives or when using stool samples due to contamination [54,55]. High concordance with the phenotypic AST and Sanger sequencing methods, the ability to detect heteroresistant strains, and the feasibility of assessing clarithromycin resistance-causing alleles retrospectively from formalin-fixed paraffin-embedded gastric tissue are a few of the most notable features of the droplet digital PCR method [23,24,56].

As conventional methods require prior knowledge of mutations to use specific primers for detecting them in only a limited number of target genes, there is a risk of missing novel, rare, or complex antibiotic resistance mechanisms. Consequently, these techniques permit the detection of resistance mutations at a restricted number of sites. However, due to *H. pylori*’s vast genetic diversity and high mutation and recombination rates, the mechanisms of antimicrobial resistance are heterogeneous and complex especially for antibiotics such as metronidazole and amoxicillin. This is particularly important in areas with high clarithromycin resistance [24,57]. Using metronidazole as an example, mutations or DNA sequence alterations associated with resistance have been described in multiple genes and regions, including the rdxA, frxA, fdxB, and recA genes associated with various nucleotide alterations including nonsense, missense, frameshift, deletion, and insertion. As a result, traditional PCR-based assays based on hybridization or enzymatic analytic techniques fall short of the mark when it comes to detecting complicated and new mutations or assessing resistance to several antibiotics [58].

NGS has evolved into a potent, quick (turnaround time of 24–72 h), and cost-effective technique for predicting antibiotic resistance and evaluating several genes at the same time [19,59]. These methods could be combined with bacterial culture for whole-genome sequencing (WGS) or with other molecular techniques such as PCR for deep-amplicon sequencing. NGS-based approaches enable a comprehensive view of bacterial genotypes and can simultaneously identify DNA mutations or variants of *H. pylori* genes responsible for resistance in all the commonly used antibiotics for *H. pylori* eradication, with the possibility of identifying novel or rare resistance mechanisms. With evolving data on the molecular basis of resistance, the clinical relevance of new mutations identified by these approaches can be ascertained by retrospective analysis of WGS data, in contrast to phenotypic methods that require retesting using stored samples, albeit with reduced ability to grow in vivo, or PCR-based assays that require redesigning samples. It is also capable of detecting HR and can be performed on fresh, frozen, or archived formalin-fixed paraffin-embedded gastric tissue blocks [19,59,60,61]. Several NGS-based technologies are even affordable in low-income countries, and the high concordance between mutations and phenotypic drug resistance makes them a desirable option for monitoring genotypic markers of antibiotic resistance in *H. pylori* [23,24,59,62]. The combination of WGS and phenotypic methodologies, according to some experts, provides a more comprehensive approach to antimicrobial resistance surveillance [63].

In contrast to deep-amplicon sequencing, WGS approaches require prior culturing of *H. pylori*, which is a significant limitation. Furthermore, the presence of fragmented DNA makes WGS more challenging when applied to formalin-fixed, paraffin-embedded tissue. In comparison to WGS, the PCR-based targeted sequencing approach using NGS is more practicable and cost-effective in routine clinical practice [64]. Hulten et al. recently compared targeted NGS and culture-based *H. pylori* susceptibility testing using clinical isolates and paired formalin-fixed, paraffin-embedded gastric biopsies. NGS showed good agreement with the agar dilution method for clarithromycin (k = 0.90012, *p =* 0.0001) and levofloxacin (k = 0.78161, *p =* 0.0001) when applied to clinical isolates but less satisfactory agreement for metronidazole (k = 0.5588, *p =* 0.0001) and amoxicillin (k = 0.21400, *p* = 0.0051). For clarithromycin, levofloxacin, metronidazole, and amoxicillin, the accuracy of NGS in predicting resistance in clinical isolates was 97.1%, 89.5%, 77.6%, and 95.9%, respectively. When carried out on formalin-fixed, paraffin-embedded gastric biopsies, NGS showed good agreement with agar dilution for clarithromycin (k = 0.81236, *p =* 0.0001) and levofloxacin (k = 0.74953, *p* = 0.0001) but less satisfactory agreement for metronidazole (k = 0.54645, *p =* 0.0001) and amoxicillin (k = 0.21400, *p* = 0.0051). NGS was able to predict resistance in formalin-fixed, paraffin-embedded biopsies with an accuracy of 94.1% for clarithromycin, 87.7% for levofloxacin, 77.7% for metronidazole, and 95.9% for amoxicillin. These results are significant as it eliminates the need for additional specimens and cultures. However, the results for amoxicillin and metronidazole are relatively poor, most likely due to the insufficient molecular characterization of these antibiotic resistance mechanisms as explained above [58]. To obtain a perfect correlation with phenotypic data, a comprehensive WGS-based analysis may aid in identifying additional mutations of these genes or of others or other resistance mechanisms. Moss et al. were the first to report the accuracy of NGS in determining the *H. pylori* antimicrobial susceptibility profile in stool samples, finding that the results obtained from stool samples were concordant with those obtained from FFPE gastric biopsies in 91.4% of cases and fresh gastric specimens in 92.2% of cases. Clarithromycin (k = 0.94), levofloxacin (k = 0.88), and metronidazole (k = 0.89) had good agreement between stool and fresh stomach samples. This is a huge step forward in obtaining antibiotic susceptibility data using NGS without the discomfort, costs, and risks associated with endoscopy [65].

NGS-based techniques are incapable of determining MIC, and the contribution of each mutation to MIC is not always obvious. Moreover, as with other genomic-based methods, phenotypic and culture-based susceptibility testing and genomic prediction of resistance do not always correspond. NGS can also underestimate resistance caused by non-genetic mechanisms [19,24,59,66]. There is a need for additional research to determine which genes or gene involvement patterns correlate most strongly with outcomes [58,66]. To overcome the diminished predictive value that could be the result of DNA fragmentation in FFPE tissue samples or associated contamination in stool samples, excellent DNA extraction methods need to be developed. Consequently, studies with broader gene coverages, larger sample sizes, or multicenter designs from various geographic regions are necessary. Standardized and user-friendly computational software and tools must be created so that NGS data may be easily analyzed and implemented in routine clinical settings [24,64].

## 5. Empirical Therapy vs. Susceptibility-Guided Tailored Therapy

According to the principles of antibiotic stewardship, susceptibility assays (culture or molecular-based) have been proposed for any bacterial infection, and *H. pylori* treatment should not be an exception. Still, eradication protocols are mainly empirical in the majority of settings. The recent Maastricht VI/Florence consensus report, while recommending AST prior to prescribing first-line therapy as a plausible approach, acknowledges its limited utility in real-world settings [67]. Several meta-analyses compared cure rates for first-line therapy using an AST-based vs. an empirical method [68,69]. The evidence suggests that although the AST-based approach performs better when using triple-drug therapy, the difference is not significant when the most recent quadruple regimens are employed. The majority of international guidelines recommend avoiding clarithromycin and levofloxacin when local resistance patterns indicate rates of >15%. As a result of the fact that worldwide data reveal this to be the case in the majority of regions throughout the world, the treatment is already being shifted to bismuth-based quadruple therapy [19].

A recent AGA clinical practice update on the management of refractory *H. pylori* recommended susceptibility testing to guide the selection of subsequent regimens after two failed eradication attempts with confirmed patient adherence [11]. This is in accordance with Maastricht V’s recommendations [70]. However, in real-world contexts, this situation is complicated by logistical difficulties in acquiring *H. pylori* resistance profiles and a lack of solid evidence favoring tailored therapy over empirical therapy in patients whose *H. pylori* treatment has failed [1,71]. Several meta-analyses, including a recent update, have found comparable *H. pylori* cure rates between susceptibility-guided tailored therapies and empirical therapy for second-line treatment [72,73,74,75,76]. Recent data from the European Registry for the Management of *Helicobacter pylori* (Hp-EuReg) indicate that eradication rates of more than 90% can be achieved using various empirical regimens as second-line therapy [77]. Furthermore, patients who have had two or more failed *H. pylori* medications demonstrated equal eradication rates across customized and empirical therapies [74,75,76,78,79].

Several studies have given contradictory results about the cost-effectiveness of susceptibility-guided treatment [71]. Moreover, even when the susceptibility profile of *H. pylori* is known, eradication rates are not 100 percent, indicating a disparity between in vitro susceptibility and in vivo eradication [71,76]. A recent systematic review of clinical studies that investigated the efficacy of second-line treatments revealed a cure rate of only 72% in patients harboring a clarithromycin-susceptible strain after prior clarithromycin treatment [80]. Consequently, there is insufficient evidence to support the routine use of susceptibility-guided, individualized therapy after *H. pylori* eradication failure at this time. Recently, the Maastricht VI Consensus suggested that second-line and rescue therapies should be guided by local resistance patterns as determined by susceptibility testing and monitoring eradication rates to maximize treatment success [1]. Therefore, there is an immediate need for a well-designed trial comparing AST-directed rescue therapy with empirical rescue therapy. With the advent of molecular techniques, *H. pylori* management can now enter the realm of antimicrobial stewardship. In addition, the development of non-invasive molecular techniques may, in the near future, remove many rate-limiting obstacles to the pervasive use of an AST-guided approach. On the other hand, further research is needed to evaluate the efficacy of new empirical medicines, such as vonoprazan-based therapies, with susceptibility-guided therapy. Nonetheless, susceptibility tests should be performed routinely, even before prescribing first-line treatment, in specialized centers with an interest in *H. pylori* management.

## 6. Conclusions

Antimicrobial resistance is a ubiquitous problem that necessitates a variety of solutions. Drug-resistant *H. pylori* is becoming an increasing health problem as a result of a considerable drop in the eradication rate of empirical therapy, which could ultimately lead to clinical complications associated with persistent infection as well as increased healthcare expenses. Therefore, periodic regional drug susceptibility surveillance programs and the implementation of policies for the prudent use of antibiotics should be of the utmost importance. Due to the advancement of molecular methods for detecting antibiotic resistance, the treatment of *H. pylori* can now enter the domain of antimicrobial stewardship. The use of genomics to investigate antibiotic resistance in bacteria of public health concern is a burgeoning field of study; the combination of WGS and traditional phenotypic resistance data can provide powerful results in discovering new resistance mechanisms. In addition, the development of non-invasive molecular techniques may in the near future remove many rate-limiting barriers to the widespread application of an AST-guided approach. Although a susceptibility-based strategy should be used whenever possible, empirical therapy based on prior medication history and local resistance patterns is an acceptable alternative when logistical and financial factors are considered. There is a need for the development of novel non-invasive antibiotic susceptibility testing tools and the standardization of existing protocols. Future research is required to determine the relative contribution of each mutation to MIC and the correlation between gene involvement patterns and outcomes.

## Figures and Tables

**Figure 1 diagnostics-13-02944-f001:**
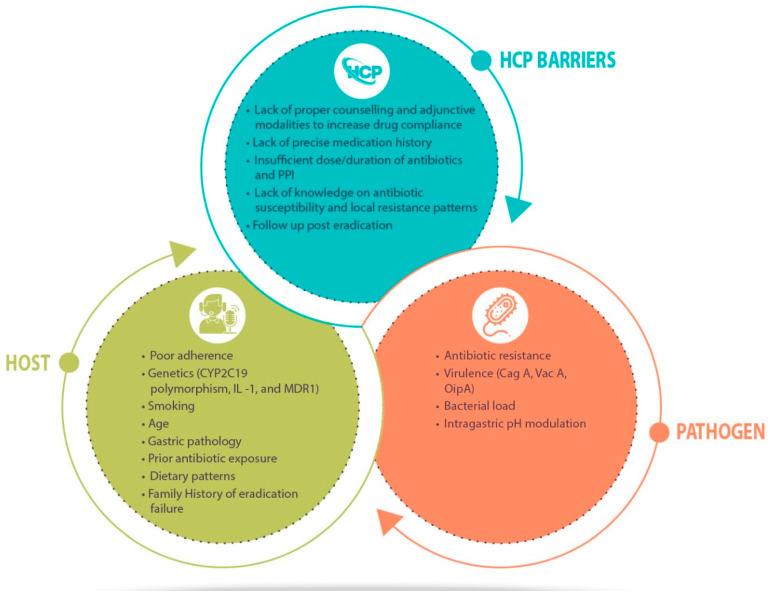
Factors responsible for *Helicobacter pylori* eradication failure.

**Figure 2 diagnostics-13-02944-f002:**
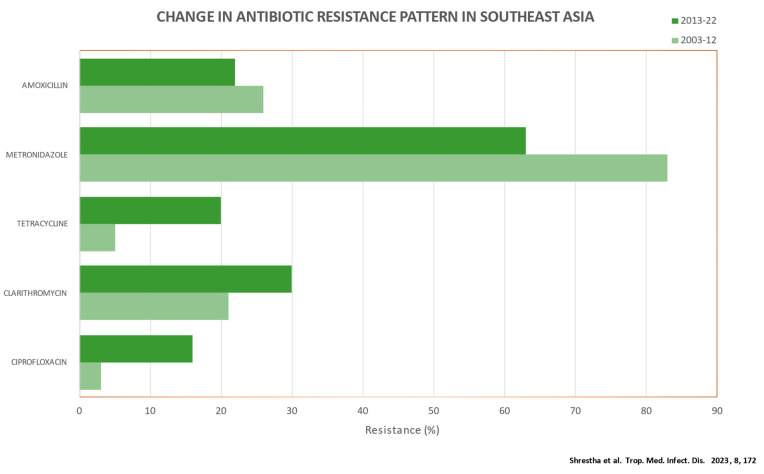
Decade-wise trends of antibiotic resistance among *H. pylori* strains (2003–2012 vs. 2013–2022) [15].

**Figure 3 diagnostics-13-02944-f003:**
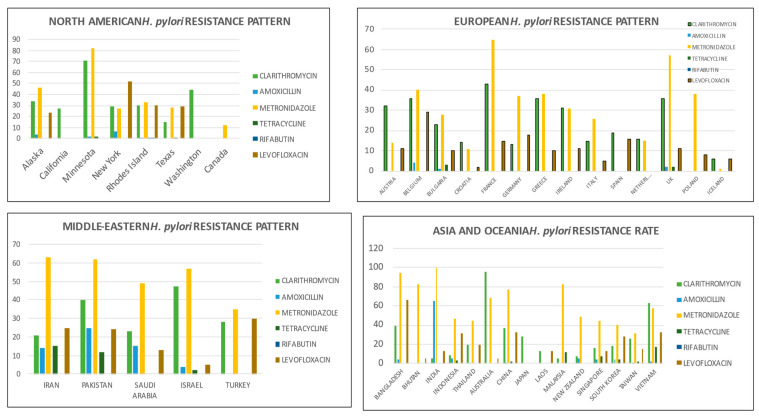
Region-wise data of antibiotic resistance among *H. pylori* strains.

**Figure 4 diagnostics-13-02944-f004:**
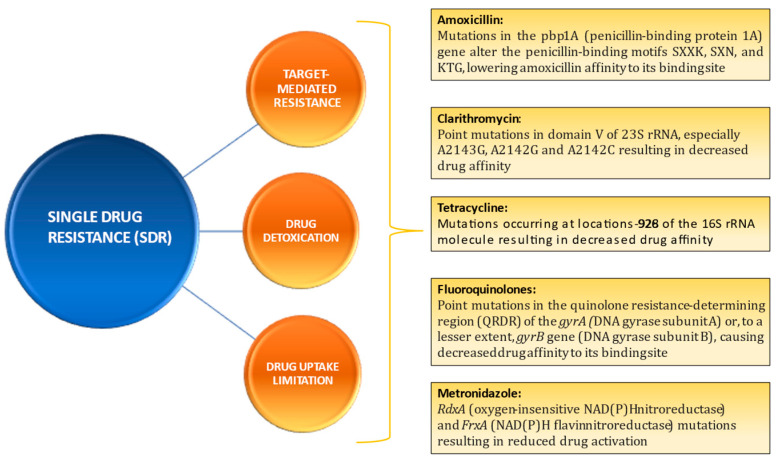
Predominant molecular mechanisms of single-drug resistance in *Helicobacter pylori*.

**Figure 5 diagnostics-13-02944-f005:**
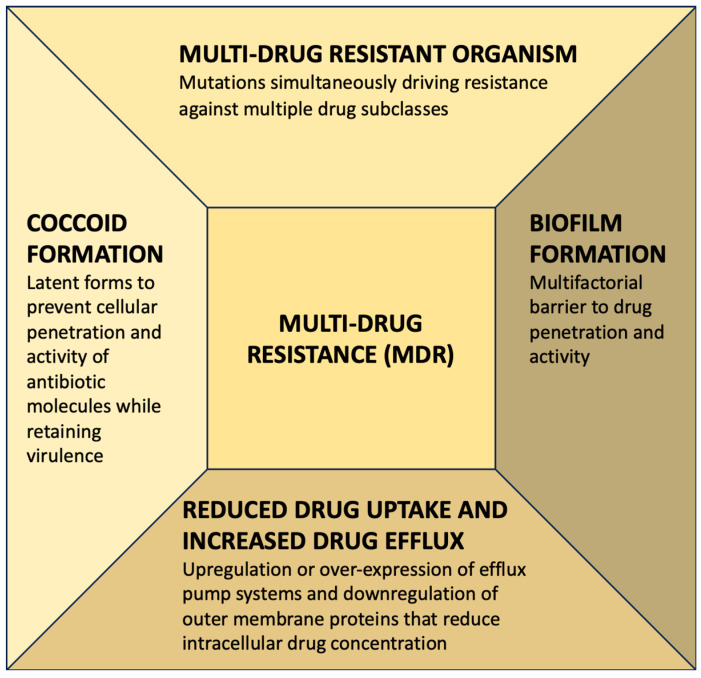
Predominant mechanisms attributable to *H. pylori* multidrug resistance.

**Figure 6 diagnostics-13-02944-f006:**
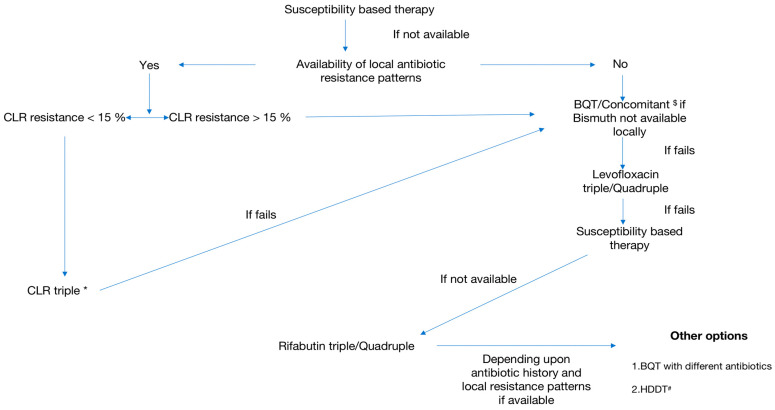
Treatment algorithm for *H pylori* infection. CLR: Clarithromycin, BQT: Bismuth Quadruple Therapy, HDDT: High-dose PPI-Amoxicillin dual therapy, Levofloxacin Quadruple: Bismuth plus Levofloxacin triple therapy, Rifabutin Quadruple: Rifabutin triple plus Bismuth therapy. * High-dose PPI twice daily increases the efficacy of triple therapy [1]. # Potassium-competitive acid blocker (Vonoprazan)-based dual therapy is another option when available. $ In populations with high metronidazole resistance, a high dose of metronidazole can increase the eradication rates of *H. pylori* infection [10].

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
