# Peer review of "Drug-Resistant Helicobacter pylori: Diagnosis and Evidence-Based Approach"

_diagnostics, 2023, doi:10.3390/diagnostics13182944_

Round 1
Reviewer 1 Report
The article: “Diagnosis of Drug Resistance Helicobacter pylori an update” has been reviewed. Vaneet Jearth et al. have reported a review about treatment failure and antibiotic resistance mechanisms of Helicobacter pylori (H. pylori), analyzing the current diagnostic strategies. I really appreciate the author’s efforts, however, after careful consideration and evaluation, I don’t believe that it is publishable in its current form. The main reasons are summarized as following:
- A more defined scope would have enhanced the value and relevance of the review. It is unclear what the paper contributes to the existing academic literature, because it doesn't provide remarkable information in this field
- The article would have benefitted from a more rigorous critical analysis of the reviewed studies. I encourage the author to consider incorporating a more thorough evaluation of the methodology, limitations, and potential biases of the studies included in the review
I suggest the author addressing these concerns and submitting an improved version of your manuscript.
The article requires substantial rewriting and language editing to enhance its clarity and overall readability. The writing style would benefit from being more concise, cohesive, and engaging to effectively convey your research findings and arguments. There are some typographical errors present throughout the document, such as misspelled words, inconsistent punctuation, and formatting inconsistencies.
Author Response
We thank you and the Reviewers for constructive comments on our manuscript. Please see our point-by-point responses to Reviewer comments below.
The article: “Diagnosis of Drug Resistance Helicobacter pylori an update” has been reviewed. Vaneet Jearth et al. have reported a review about treatment failure and antibiotic resistance mechanisms of Helicobacter pylori (H. pylori), analyzing the current diagnostic strategies. I really appreciate the author’s efforts, however, after careful consideration and evaluation, I don’t believe that it is publishable in its current form. The main reasons are summarized as following:
- A more defined scope would have enhanced the value and relevance of the review. It is unclear what the paper contributes to the existing academic literature, because it doesn't provide remarkable information in this field.
Ans: Thank you for reviewing our paper critically. Rising antibiotic resistance among H. pylori has important downstream consequences, as highlighted in our work. We attempted to give an updated literature review on global prevalence, mechanisms of antibiotic resistance, as well as its clinical consequences and methods for detection, within the context of the journal's need to focus on the diagnosis of drug resistance in H. pylori. We attempted to incorporate relevant literature for different phenotypic and genotypic methods for detecting antibiotic resistance from the standpoint of a gastroenterologist while also highlighting current evidence for AST guided customized vs empirical eradication therapy. In addition, we suggested an approach for evidence-based management of H. pylori in both first line and rescue settings. We believe that this research will assist readers understand current improvements in H. pylori resistance detection and the necessity to change from a hopeful (empirical) approach to more specific tailored therapy using molecular approaches where available. We have compiled key insights related to drug resistance from all significant recent international guidelines under one title. Also, we have briefly highlighted other factors implicated in treatment failure apart from drug resistance in Figure 1. To better illustrate the specifics of resistance mechanisms, Figures 4 and 5 have been included.
- The article would have benefitted from a more rigorous critical analysis of the reviewed studies. I encourage the author to consider incorporating a more thorough evaluation of the methodology, limitations, and potential biases of the studies included in the review.
I suggest the author addressing these concerns and submitting an improved version of your manuscript.
Ans: Thank you for your thoughtful comments. We made suggested improvements to the amended draft, including expanding the various sections including molecular approaches with descriptions of the advantages and disadvantages of each method. In addition, we have included some new evidence published recently related to our review and added references have been highlighted.
Comments on the Quality of English Language
The article requires substantial rewriting and language editing to enhance its clarity and overall readability. The writing style would benefit from being more concise, cohesive, and engaging to effectively convey your research findings and arguments. There are some typographical errors present throughout the document, such as misspelled words, inconsistent punctuation, and formatting inconsistencies.
Ans: We have revised the entire paper for grammatical errors and inconsistencies. We've additionally paraphrased most part of our paper to make our point more obvious.
Reviewer 2 Report
The manuscript is devoted to the problem of antibiotic resistance of one of the most dangerous human pathogens - Helicobacter pylori. Mechanisms of resistance to antibiotics, methods of their detection, as well as the algorithm of the most appropriate treatment in each case have been described. The manuscript is suitable for publication, but there are a few remarks about the figures.
1. Figure 2. It is worth placing bars with different decades next to each other, and not on top of each other, because it is difficult to assess the increase or decrease in resistance to each antibiotic, and it also creates the false impression that resistance to metronidazole can exceed 100%.
2. Figure 4. Arrows in the flowchart should not be so bold. You should also avoid non-horizontal and non-vertical arrows (at an angle), it is better to replace them with angular arrows
Author Response
We thank you and the Reviewers for constructive comments on our manuscript. Please see our point-by-point responses to Reviewer comments below.
The manuscript is devoted to the problem of antibiotic resistance of one of the most dangerous human pathogens - Helicobacter pylori. Mechanisms of resistance to antibiotics, methods of their detection, as well as the algorithm of the most appropriate treatment in each case have been described. The manuscript is suitable for publication, but there are a few remarks about the figures.
- Figure 2. It is worth placing bars with different decades next to each other, and not on top of each other, because it is difficult to assess the increase or decrease in resistance to each antibiotic, and it also creates the false impression that resistance to metronidazole can exceed 100%.
Ans: Thank you for your feedback on our manuscript. We have incorporated the suggested edits into the revised manuscript.
- Figure 4. Arrows in the flowchart should not be so bold. You should also avoid non-horizontal and non-vertical arrows (at an angle), it is better to replace them with angular arrows.
Ans: Thank you for your important feedback. We have made the suggested changes to figure 4.
Reviewer 3 Report
The text contains several errors in writing the abbreviated name of Helicobacter pylori that should be corrected (pay attention to the italics).
Antibiotic abbreviations in Figures 2 and 3 should be explained below the figure so that they are self-explanatory.
Author Response
We thank you and the Reviewers for constructive comments on our manuscript. Please see our point-by-point responses to Reviewer comments below.
The text contains several errors in writing the abbreviated name of Helicobacter pylori that should be corrected (pay attention to the italics).
Ans: Thank you for critical review of our manuscript. We have made the suggested changes in whole paper.
Antibiotic abbreviations in Figures 2 and 3 should be explained below the figure so that they are self-explanatory.
Ans: Thank you for this important suggestion. We have made the suggested changes in our revised draft and included full name of antibiotics in our figures.